# MelNet: A Generative Model for Audio in the Frequency Domain

## Abstract

Capturing high-level structure in audio waveforms is challenging because a single second of audio spans tens of thousands of timesteps. While long-range dependencies are difficult to model directly in the time domain, we show that they can be more tractably modelled in two-dimensional time-frequency representations such as spectrograms. By leveraging this representational advantage, in conjunction with a highly expressive probabilistic model and a multiscale generation procedure, we design a model capable of generating high-fidelity audio samples which capture structure at timescales which time-domain models have yet to achieve. We demonstrate that our model captures longer-range dependencies than time-domain models such as WaveNet across a diverse set of unconditional generation tasks, including single-speaker speech generation, multi-speaker speech generation, and music generation.

## 1 Introduction

Audio waveforms have complex structure at drastically varying timescales, which presents a challenge for generative models. Local structure must be captured to produce high-fidelity audio, while long-range dependencies spanning tens of thousands of timesteps must be captured to generate audio which is globally consistent. Existing generative models of waveforms such as WaveNet (van den Oord et al., 2016a) and SampleRNN (Mehri et al., 2016) are well-adapted to model local dependencies, but as these models typically only backpropagate through a fraction of a second, they are unable to capture high-level structure that emerges on the scale of several seconds.

We introduce a generative model for audio which captures longer-range dependencies than existing end-to-end models. We primarily achieve this by modelling 2D time-frequency representations such as spectrograms rather than 1D time-domain waveforms (Figure 1). The temporal axis of a spectrogram is orders of magnitude more compact than that of a waveform, meaning dependencies that span tens of thousands of timesteps in waveforms only span hundreds of timesteps in spectrograms. In practice, this enables our spectrogram models to generate unconditional speech and music samples with consistency over multiple seconds whereas time-domain models must be conditioned on intermediate features to capture structure at similar timescales.

Modelling spectrograms can simplify the task of capturing global structure, but can weaken a model's ability to capture local characteristics that correlate with audio fidelity. Producing high-fidelity audio has been challenging for existing spectrogram models, which we attribute to the lossy nature of spectrograms and oversmoothing artifacts which result from insufficiently expressive models. To reduce information loss, we model high-resolution spectrograms which have the same dimensionality as their corresponding time-domain signals. To limit oversmoothing, we use a highly expressive autoregressive model which factorizes the distribution over both the time and frequency dimensions.

Modelling both fine-grained details and high-level structure in high-dimensional distributions is known to be challenging for autoregressive models. To capture both local and global structure in spectrograms with hundreds of thousands of dimensions, we employ a multiscale approach which generates spectrograms in a coarse-to-fine manner. A low-resolution, subsampled spectrogram that captures high-level structure is generated initially, followed by an iterative upsampling procedure that adds high-resolution details.

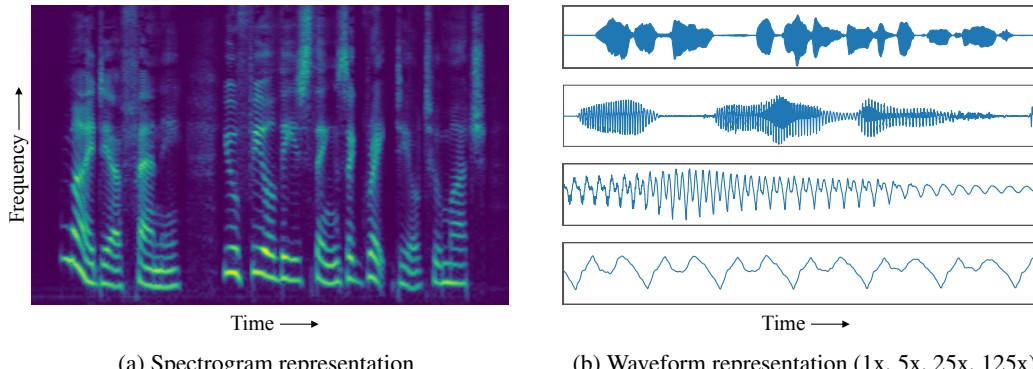

<table>
<tr><td>(a) Spectrogram representation</td><td>(b) Waveform representation (1x, 5x, 25x, 125x)</td></tr>
</table>

Figure 1: Spectrogram and waveform representations of the same 4 second audio signal. The waveform spans nearly 100,000 timesteps whereas the temporal axis of the spectrogram spans roughly 400. Complex structure is nested within the temporal axis of the waveform at various timescales, whereas the spectrogram has structure which is smoothly spread across the time-frequency plane.

Combining these representational and modelling techniques yields a highly expressive and broadly applicable generative model of audio. Our contributions are are as follows:

- We introduce MelNet, a generative model for spectrograms which couples a fine-grained autoregressive model and a multiscale generation procedure to jointly capture local and global structure.

- We show that MelNet is able to model longer-range dependencies than existing time-domain models. Additionally, we include an ablation to demonstrate that multiscale modelling is essential for modelling long-range dependencies.

- We demonstrate that MelNet is broadly applicable to a variety of audio generation tasks, including unconditional speech and music generation. Furthermore, MelNet is able to model highly multimodal data such as multi-speaker and multilingual speech.

## 2 PRELIMINARIES

We briefly present background regarding spectral representations of audio. Audio is represented digitally as a one-dimensional, discrete-time signal $y = (y_1, \ldots, y_n)$. Existing generative models for audio have predominantly focused on modelling these time-domain signals directly. We instead model spectrograms, which are two-dimensional time-frequency representations which contain information about how the frequency content of an audio signal varies through time. Spectrograms are computed by taking the squared magnitude of the short-time Fourier transform (STFT) of a time-domain signal, i.e. $x = \|\text{STFT}(y)\|^2$. The value of $x_{ij}$ (referred to as amplitude or energy) corresponds to the squared magnitude of the $j$th element of the frequency response at timestep $i$. Each slice $x_{i,*}$ is referred to as a *frame*. We assume a time-major ordering, but following convention, all figures are displayed transposed and with the frequency axis inverted.

Time-frequency representations such as spectrograms highlight how the tones and pitches within an audio signal vary through time. Such representations are closely aligned with how humans perceive audio. To further align these representations with human perception, we convert the frequency axis to the Mel scale and apply an elementwise logarithmic rescaling of the amplitudes. Roughly speaking, the Mel transformation aligns the frequency axis with human perception of pitch and the logarithmic rescaling aligns the amplitude axis with human perception of loudness.

Spectrograms are lossy representations of their corresponding time-domain signals. The Mel transformation discards frequency information and the removal of the STFT phase discards temporal information. When recovering a time-domain signal from a spectrogram, this information loss manifests as distortion in the recovered signal. To minimize these artifacts and improve the fidelity of generated audio, we model high-resolution spectrograms. The temporal resolution of a spectrogram can be increased by decreasing the STFT hop size, and the frequency resolution can be increased by

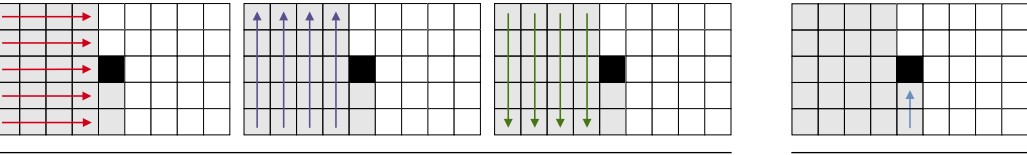

(a) Time-delayed stack                                   (b) Frequency-delayed stack

Figure 2: The context $x_{<ij}$ (grey) for the element $x_{ij}$ (black) is encoded using 4 RNNs. Three of these are used in the time-delayed stack to extract features from preceding frames. The fourth is used in the frequency-delayed stack to extract features from all preceding elements within the current frame. Each arrow denotes an individual RNN cell and arrows of the same color use shared parameters.

increasing the number of Mel channels. Generated spectrograms are converted back to time-domain signals using classical spectrogram inversion algorithms. We experiment with both Griffin-Lim (Griffin & Lim, 1984) and a gradient-based inversion algorithm (Decorsière et al., 2015), and ultimately use the latter as it generally produced audio with fewer artifacts.

## 3 PROBABILISTIC MODEL

We use an autoregressive model which factorizes the joint distribution over a spectrogram $x$ as a product of conditional distributions. Given an ordering of the dimensions of $x$, we define the context $x_{<ij}$ as the elements of $x$ that precede $x_{ij}$. We default to a row-major ordering which proceeds through each frame $x_{i,*}$ from low to high frequency, before progressing to the next frame. The joint density is factorized as

$$p(x) = \prod_i \prod_j p(x_{ij} \mid x_{<ij};\ \theta_{ij}), \tag{1}$$

where $\theta_{ij}$ parameterizes a univariate density over $x_{ij}$. We model each factor distribution as a Gaussian mixture model with $K$ components. Thus, $\theta_{ij}$ consists of $3K$ parameters corresponding to means $\{\mu_{ijk}\}_{k=1}^K$, standard deviations $\{\sigma_{ijk}\}_{k=1}^K$, and mixture coefficients $\{\pi_{ijk}\}_{k=1}^K$. The resulting factor distribution can then be expressed as

$$p(x_{ij} \mid x_{<ij};\ \theta_{ij}) = \sum_{k=1}^K \pi_{ijk}\, \mathcal{N}(x_{ij};\ \mu_{ijk}, \sigma_{ijk}). \tag{2}$$

Following the work on Mixture Density Networks (Bishop, 1994) and their application to autoregressive models (Graves, 2013), $\theta_{ij}$ is modelled as the output of a neural network and computed as a function of the context $x_{<ij}$. Precisely, for some network $f$ with parameters $\psi$, we have $\theta_{ij} = f(x_{<ij};\ \psi)$. A maximum-likelihood estimate for the network parameters is computed by minimizing the negative log-likelihood via gradient descent.

To ensure that the network output parameterizes a valid Gaussian mixture model, the network first computes unconstrained parameters $\{\hat{\mu}_{ijk}, \hat{\sigma}_{ijk}, \hat{\pi}_{ijk}\}_{k=1}^K$ as a vector $\hat{\theta}_{ij} \in \mathbb{R}^{3K}$, and enforces constraints on $\theta_{ij}$ by applying the following transformations:

$$\mu_{ijk} = \hat{\mu}_{ijk} \tag{3}$$

$$\sigma_{ijk} = \exp(\hat{\sigma}_{ijk}) \tag{4}$$

$$\pi_{ijk} = \frac{\exp(\hat{\pi}_{ijk})}{\sum_{k=1}^K \exp(\hat{\pi}_{ijk})}\ . \tag{5}$$

These transformations ensure the standard deviations $\sigma_{ijk}$ are positive and the mixture coefficients $\pi_{ijk}$ sum to one.

## 4 NETWORK ARCHITECTURE

To model the distribution in an autoregressive manner, we design a network which computes the distribution over $x_{ij}$ as a function of the context $x_{<ij}$. The network architecture draws inspiration

from existing autoregressive models for images (Theis & Bethge, 2015; van den Oord et al., 2016c;b; Chen et al., 2017; Salimans et al., 2017; Parmar et al., 2018; Child et al., 2019). In the same way that these models estimate a distribution pixel-by-pixel over the spatial dimensions of an image, our model estimates a distribution element-by-element over the time and frequency dimensions of a spectrogram. A noteworthy distinction is that spectrograms are not invariant to translation along the frequency axis, making 2D convolution less desirable than other 2D network primitives which do not assume invariance. Utilizing multidimensional recurrence instead of 2D convolution has been shown to be beneficial when modelling spectrograms in discriminative settings (Li et al., 2016; Sainath & Li, 2016), which motivates our use of an entirely recurrent architecture.

Similar to Gated PixelCNN (van den Oord et al., 2016b), the network has multiple *stacks* of computation. These stacks extract features from different segments of the input to collectively summarize the full context $x_{<ij}$:

- The *time-delayed* stack computes features which aggregate information from all previous frames $x_{<i,*}$.
- The *frequency-delayed* stack utilizes all preceding elements within a frame, $x_{i,<j}$, as well as the outputs of the time-delayed stack, to summarize the full context $x_{<ij}$.

The stacks are connected at each layer of the network, meaning that the features generated by layer $l$ of the time-delayed stack are used as input to layer $l$ of the frequency-delayed stack. To facilitate the training of deeper networks, both stacks use residual connections (He et al., 2016). The outputs of the final layer of the frequency-delayed stack are used to compute the unconstrained parameters $\hat{\theta}$.

## 4.1 Time-Delayed Stack

The time-delayed stack utilizes multiple layers of multidimensional RNNs to extract features from $x_{<i,*}$, the two-dimensional region consisting of all frames preceding $x_{ij}$. Each multidimensional RNN is composed of three one-dimensional RNNs: one which runs forwards along the frequency axis, one which runs backwards along the frequency axis, and one which runs forwards along the time axis. Each RNN runs along each slice of a given axis, as shown in Figure 2. The output of each layer of the time-delayed stack is the concatenation of the three RNN hidden states.

We denote the function computed at layer $l$ of the time-delayed stack (three RNNs followed by concatenation) as $\mathcal{F}_l^t$. At each layer, the time-delayed stack uses the feature map from the previous layer, $h^t[l-1]$, to compute the subsequent feature map $\mathcal{F}_l^t\big(h^t[l-1]\big)$ which consists of the three concatenated RNN hidden states. When using residual connections, the computation of $h^t[l]$ from $h^t[l-1]$ becomes

$$h_{ij}^t[l] = W_l^t \mathcal{F}_l^t\big(h^t[l-1]\big)_{ij} + h_{ij}^t[l-1]. \tag{6}$$

To ensure the output $h_{ij}^t[l]$ is only a function of frames which lie in the context $x_{<ij}$, the inputs to the time-delayed stack are shifted backwards one step in time: $h_{ij}^t[0] = W_0^t x_{i-1,j}$.

## 4.2 Frequency-Delayed Stack

The frequency-delayed stack is a one-dimensional RNN which runs forward along the frequency axis. Much like existing one-dimensional autoregressive models (language models, waveform models, etc.), the frequency-delayed stack operates on a one-dimensional sequence (a single frame) and estimates the distribution for each element conditioned on all preceding elements. The primary difference is that it is also conditioned upon the outputs of the time-delayed stack, allowing it to use the full two-dimensional context $x_{<ij}$.

We denote the function computed by the frequency-delayed stack as $\mathcal{F}_l^f$. At each layer, the frequency-delayed stack takes two inputs: the the previous-layer outputs of the frequency-delayed stack, $h_{ij}^f[l-1]$, and the current-layer outputs of the time-delayed stack $h_{ij}^t[l]$. These inputs are summed and used as input to a one-dimensional RNN to produce the output feature map $\mathcal{F}_l^f\big(h^f[l-1],\ h^t[l]\big)$ which consists of the RNN hidden state:

$$h_{ij}^f[l] = W_l^f \mathcal{F}_l^f\big(h^f[l-1],\ h^t[l]\big)_{ij} + h_{ij}^f[l-1]. \tag{7}$$

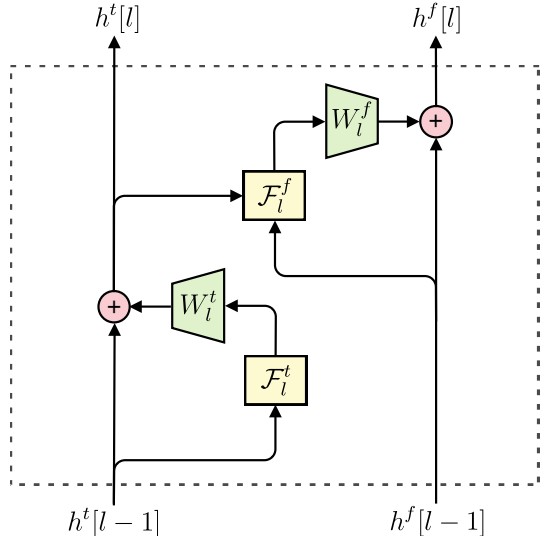

Figure 3: Computation graph for a single layer of the network. $\mathcal{F}_l^t$ and $\mathcal{F}_l^f$ are the functions computed by the time-delayed stack and frequency-delayed stack, respectively. The outputs of these functions are projected (by the matrices $W_l^t$ and $W_l^f$) and summed with the layer inputs to form residual blocks.

To ensure that $h_{ij}^f[l]$ is computed using only elements in the context $x_{<ij}$, the inputs to the frequency-delayed stack are shifted backwards one step along the frequency axis: $h_{ij}^f[0] = W_0^f x_{i,j-1}$. At the final layer, layer $L$, a linear map is applied to the output of the frequency-delayed stack to produce the unconstrained Gaussian mixture model parameters, i.e. $\hat{\theta}_{ij} = W_\theta h_{ij}^f[L]$.

### 4.3 CONDITIONING

To incorporate conditioning information into the model, conditioning features $z$ are simply projected onto the input layer along with the inputs $x$:

$$h_{ij}^t[0] = W_0^t x_{i-1,j} + W_z^t z_{ij} \tag{8}$$

$$h_{ij}^f[0] = W_0^f x_{i,j-1} + W_z^f z_{ij}. \tag{9}$$

Reshaping, upsampling, and broadcasting can be used as necessary to ensure the conditioning features have the same time and frequency shape as the input spectrogram, e.g. a one-hot vector representation for speaker ID would first be broadcast along both the time and frequency axes.

## 5 MULTISCALE MODELLING

To improve audio fidelity, we generate high-resolution spectrograms which have the same dimensionality as their corresponding time-domain representations. Under this regime, a single training example has several hundreds of thousands of dimensions. Capturing global structure in such high-dimensional distributions is challenging for autoregressive models, which are biased towards capturing local dependencies. To counteract this, we utilize a multiscale approach which effectively permutes the autoregressive ordering so that a spectrogram is generated in a coarse-to-fine order.

The elements of a spectrogram $x$ are partitioned into $G$ tiers $x^1, \ldots, x^G$, such that each successive tier contains higher-resolution information. We define $x^{<g}$ as the union of all tiers which precede $x^g$, i.e. $x^{<g} = (x^1, \ldots, x^{g-1})$. The distribution is factorized over tiers:

$$p(x; \psi) = \prod_g p(x^g \mid x^{<g}; \psi^g), \tag{10}$$

and the distribution of each tier is further factorized element-by-element as described in Section 3. We explicitly include the parameterization by $\psi = (\psi^1, \ldots, \psi^G)$ to indicate that each tier is modelled by a separate network.

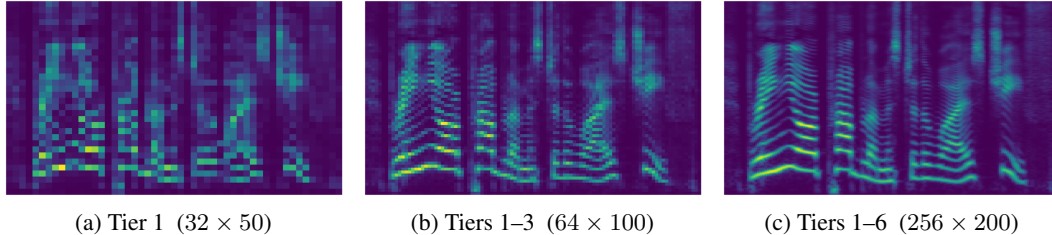

(a) Tier 1 $(32 \times 50)$     (b) Tiers 1–3 $(64 \times 100)$     (c) Tiers 1–6 $(256 \times 200)$

Figure 4: A sampled spectrogram viewed at different stages of the multiscale generation procedure. The initial tier dictates high-level structure and subsequent tiers add fine-grained details. Each upsampling tier doubles the resolution of the spectrogram.

(a) $p(x^1; \psi^1)$     (b) $p(x^2 \mid x^1; \psi^2)$     (c) $p(x^3 \mid x^1, x^2; \psi^3)$     (d) $p(x^4 \mid x^1, x^2, x^3; \psi^4)$

Figure 5: Schematic showing how tiers of the multiscale model are interleaved and used to condition the distribution for the subsequent tier. a) The initial tier is generated unconditionally. b) The second tier is generated conditionally given the the initial tier. c) The outputs of tiers 1 and 2 are interleaved along the frequency axis and used to condition the generation of tier 3. d) Tier 3 is interleaved along the time axis with all preceding tiers and used to condition the generation of tier 4.

## 5.1 TRAINING

During training, the tiers are generated by recursively partitioning a spectrogram into alternating rows along either the time or frequency axis. We define a function `split` which partitions an input into even and odd rows along a given axis. The initial step of the recursion applies the `split` function to a spectrogram $x$, or equivalently $x^{<G+1}$, so that the even-numbered rows are assigned to $x^G$ and the odd-numbered rows are assigned to $x^{<G}$. Subsequent tiers are defined similarly in a recursive manner:

$$x^g, \ x^{<g} = \texttt{split}(x^{<g+1}). \tag{11}$$

At each step of the recursion, we model the distribution $p(x^g \mid x^{<g}; \psi^g)$. The final step of the recursion models the unconditional distribution over the initial tier $p(x^1; \psi^1)$.

To model the conditional distribution $p(x^g \mid x^{<g}; \psi^g)$, the network at each tier needs a mechanism to incorporate information from the preceding tiers $x^{<g}$. To this end, we add a feature extraction network which computes features from $x^{<g}$ which are used condition the generation of $x^g$. We use a multidimensional RNN consisting of four one-dimensional RNNs which run bidirectionally along slices of both axes of the context $x^{<g}$. A layer of the feature extraction network is similar to a layer of the time-delayed stack, but since the feature extraction network is not causal, we include an RNN which runs backwards along the time axis and do not shift the inputs. The hidden states of the RNNs in the feature extraction network are used to condition the generation of $x^g$. As each tier doubles the resolution, the features extracted from $x^{<g}$ have the same time and frequency shape as $x^g$, allowing the conditioning mechanism described in section 4.3 to be used straightforwardly.

## 5.2 SAMPLING

To sample from the multiscale model we iteratively sample a value for $x^g$ conditioned on $x^{<g}$ using the learned distributions defined by the estimated network parameters $\hat{\psi} = (\hat{\psi}^1, \ldots, \hat{\psi}^G)$. The initial tier, $x^1$, is generated unconditionally by sampling from $p(x^1; \hat{\psi}^1)$ and subsequent tiers are sampled from $p(x^g \mid x^{<g}; \hat{\psi}^g)$. At each tier, the sampled $x^g$ is interleaved with the context $x^{<g}$:

$$x^{<g+1} = \texttt{interleave}(x^g, \ x^{<g}). \tag{12}$$

The `interleave` function is simply the inverse of the `split` function. Sampling terminates once a full spectrogram, $x^{<G+1}$, has been generated. A spectrogram generated by a multiscale model is shown in Figure 4 and the sampling procedure is visualized schematically in Figure 5.

## 6 EXPERIMENTS

To demonstrate the MelNet is broadly applicable as a generative model for audio, we train the model on a diverse set of audio generation tasks (single-speaker speech generation, multi-speaker speech generation, and music generation) using three publicly available datasets. Generated audio samples for each task are available on the accompanying web page https://audio-samples.github.io. We include samples generated using the priming and biasing procedures described by Graves (2013). Biasing lowers the temperature of the predictive distribution and priming seeds the model state with a given sequence of audio prior to sampling. Hyperparameters for all experiments are available in Appendix A.

Speech and music have rich hierarchies of latent structure. Speech has complex linguistic structure (phonemes, words, syntax, semantics, etc.) and music has highly compositional musical structure (notes, chords, melody and rhythm, etc.). The presence of these latent structures in generated samples can be used as a proxy for how well a generative model has learned dependencies at various timescales. As such, a qualitative analysis of unconditional samples is an insightful method of evaluating generative models of audio. To facilitate such a qualitative evaluation, we train MelNet on each of the three unconditional generation tasks and include samples on the accompanying web page. For completeness, we briefly provide some of our own qualitative observations regarding the generated samples (Sections 6.1, 6.2, and 6.3). In addition to qualitative analysis, we conduct a human evaluation experiment to quantitatively compare how well WaveNet and MelNet capture high-level structure (Section 6.4). Lastly, we ablate the impact of the multiscale generation procedure on MelNet's ability model long-range dependencies (Section 6.5).

### 6.1 SINGLE-SPEAKER SPEECH

To test MelNet's ability to model a single speaker in a controlled environment, we utilize the Blizzard 2013 dataset (King, 2011), which consists of audiobook narration performed in a highly animated manner by a professional speaker. We find that MelNet frequently generates samples that contain coherent words and phrases. Even when the model generates incoherent speech, the intonation, prosody, and speaking style remain consistent throughout the duration of the sample. Furthermore, the model learns to produce speech using a variety of character voices and learns to generate samples which contain elements of narration and dialogue. Biased samples tend to contain longer strings of comprehensible words but are read in a less expressive fashion. When primed with a real sequence of audio, MelNet is able to continue sampling speech which has consistent speaking style and intonation.

### 6.2 MULTI-SPEAKER SPEECH

Audiobook data is recorded in a highly controlled environment. To demonstrate MelNet's capacity to model distributions with significantly more variation, we utilize the VoxCeleb2 dataset (Chung et al., 2018). The VoxCeleb2 dataset consists of over 2,000 hours of speech data captured with real world noise including laughter, cross-talk, channel effects, music and other sounds. The dataset is also multilingual, with speech from speakers of 145 different nationalities, covering a wide range of accents, ages, ethnicities and languages. When trained on the VoxCeleb2 dataset, we find that MelNet is able to generate unconditional samples with significant variation in both speaker characteristics (accent, language, prosody, speaking style) as well as acoustic conditions (background noise and recording quality). While the generated speech is often not comprehensible, samples can often be identified as belonging to a specific language, indicating that the model has learned distinct modalities for different languages. Furthermore, it is difficult to distinguish real and fake samples which are spoken in foreign languages. For foreign languages, semantic structures are not understood by the listener and cannot be used to discriminate between real and fake. Consequently, the listener must rely largely on phonetic structure, which MelNet is able to realistically model.

### 6.3 MUSIC

To show that MelNet can model audio modalities other than speech, we apply the model to the task of unconditional music generation. We utilize the MAESTRO dataset (Hawthorne et al., 2018), which consists of over 172 hours of solo piano performances. The samples demonstrate that MelNet learns musical structures such as melody and harmony. Furthermore, generated samples often maintain consistent tempo and contain interesting variation in volume, timbre, and rhythm.

|  | WaveNet | MelNet |
|---|---|---|
| Blizzard | 0.0 % | 100.0 % |
| VoxCeleb2 | 0.0 % | 100.0 % |
| MAESTRO | 4.2 % | 95.8 % |

(a) Comparison between MelNet and WaveNet. Both models are trained in an entirely unsupervised manner.

|  | Wave2Midi2Wave | MelNet |
|---|---|---|
| MAESTRO | 37.7 % | 62.3 % |

(b) Comparison between MelNet and Wave2Midi2Wave. Wave2Midi2Wave is a two-stage model consisting of a Music Transformer trained on labelled MIDI followed by a conditional WaveNet model. The MelNet model, on the other hand, is trained without any intermediate supervision.

Table 1: Selection rates of human evaluators when asked to identify which model generates samples with longer-term structure. Results show that MelNet captures long-range structure better than WaveNet. Furthermore, MelNet outperforms a two-stage model which conditions WaveNet on generated MIDI.

## 6.4 HUMAN EVALUATION

Making quantitative comparisons with existing generative models such as WaveNet is difficult for various reasons and previous works have ultimately relied on largely empirical evaluations by the reader (Dieleman et al., 2018). To allow the reader to make these judgements for themselves, we provide samples from both WaveNet and MelNet for each of the tasks described in the previous sections. Furthermore, in an effort to provide quantitative metrics to support the claim that MelNet generates samples with improved long-range structure in comparison to WaveNet, we conduct a human experiment whereby participants are presented anonymized samples from both models and asked to select which sample exhibits longer-term structure. We resort to such evaluations since standard metrics for evaluation of generative models such as density estimates cannot be used to compare WaveNet and MelNet as that these models operate on different representations.

The methodology for this experiment is as follows. For each of the three unconditional audio generation tasks, we generated 50 samples from WaveNet and 50 samples from MelNet. Participants were shown an anonymized, randomly-drawn sample from each model and instructed to "select the sample which has more coherent long-term structure." We collected 50 evaluations for each task. Results, shown in Table 1a, show that evaluators overwhelmingly agreed that samples generated by MelNet had more coherent long-range structure than samples from WaveNet across all tasks.

In addition to comparing MelNet to an unconditional WaveNet model for music generation, we also compare to a two-stage Wave2Midi2Wave model (Hawthorne et al., 2018) which conditions WaveNet on MIDI generated by a separately-trained Music Transformer (Huang et al., 2018). The two-stage Wave2Midi2Wave model has the advantage of directly modelling labelled musical notes which distill much of the salient, high-level structure in music into a compact symbolic representation. Despite this, as shown by the results in Table 1b, the two-stage model does not capture long-range structure as well as a MelNet model that is trained without access to any intermediate representations.

## 6.5 ABLATION: MULTISCALE MODELLING

To isolate the impact of multiscale modelling procedure described in Section 5, we train models with varying numbers of tiers and evaluate the long-term coherence of their respective samples. As noted before, long-term coherence is difficult to quantify and we provide samples on the accompanying web page so that the reader can make their own judgements. We believe the samples clearly demonstrate that increasing the number of tiers results in samples with more coherent high-level structure. We note that our experiment varies the number of tiers from two to five. Training a single-tier model on full-resolution spectrograms was prohibitively expensive in terms of memory consumption. This highlights another benefit of multiscale modelling—large, deep networks can be allocated to learning complex distributional structure in the initial tiers while shallower networks can be used for modelling the relatively simple, low-entropy distributions in the upsampling tiers. This allows multiscale models to effectively allocate network capacity in proportion to the complexity of the modelling task.

# 7 RELATED WORK

The predominant line of research regarding generative models for audio has been directed towards modelling time-domain waveforms with autoregressive models (van den Oord et al., 2016a; Mehri et al., 2016; Kalchbrenner et al., 2018). WaveNet is a competitive baseline for audio generation, and as such, is used for comparison in many of our experiments. However, we note that the contribution of our work is in many ways complementary to that of WaveNet. MelNet is more proficient at capturing high-level structure, whereas WaveNet is capable of producing higher-fidelity audio. Several works have demonstrated that time-domain models can be used to invert spectral representations to high-fidelity audio (Shen et al., 2018; Prenger et al., 2019; Arık et al., 2019), suggesting that MelNet could be used in concert with time-domain models such as WaveNet.

Dieleman et al. (2018) and van den Oord et al. (2017) capture long-range dependencies in waveforms by utilizing a hierarchy of autoencoders. This approach requires multiple stages of models which must be trained sequentially, whereas the multiscale approach in this work can be parallelized over tiers. Additionally, these approaches do not directly optimize the data likelihood, nor do they admit tractable marginalization over the latent codes. We also note that the modelling techniques devised in these works can be broadly applied to autoregressive models such as ours, making their contributions largely complementary to ours.

Recent works have used generative adversarial networks (GANs) (Goodfellow et al., 2014) to model both waveforms and spectral representations (Donahue et al., 2018; Engel et al., 2018). As with image generation, it remains unclear whether GANs capture all modes of the data distribution. Furthermore, these approaches are restricted to generating fixed-duration segments of audio, which precludes their usage in many audio generation tasks.

Generating spectral representations is common practice for end-to-end text-to-speech models (Ping et al., 2017; Sotelo et al., 2017; Wang et al., 2017; Taigman et al., 2018). However, these models use probabilistic models which are much less expressive than the fine-grained autoregressive model used by MelNet. Consequently, these models are unsuitable for modelling high-entropy, multimodal distributions such as those involved in tasks like unconditional music generation.

The network architecture used for MelNet is heavily influenced by recent advancements in deep autoregressive models for images. Theis & Bethge (2015) introduced an LSTM architecture for autoregressive modelling of 2D images and van den Oord et al. (2016c) introduced PixelRNN and PixelCNN and scaled up the models to handle the modelling of natural images. Subsequent works in autoregressive image modelling have steadily improved state-of-the-art for image density estimation (van den Oord et al., 2016b; Salimans et al., 2017; Parmar et al., 2018; Chen et al., 2017; Child et al., 2019). We draw inspiration from many of these models, and ultimately design a recurrent architecture of our own which is suitable for modelling spectrograms rather than images. We note that our choice of architecture is not a fundamental contribution of this work. While we have designed the architecture particularly for modelling spectrograms, we did not experimentally validate whether it outperforms existing architectures and make no such claims to this effect.

We use a multidimensional recurrence in both the time-delayed stack and the upsampling tiers to extract features from two-dimensional inputs. Our multidimensional recurrence is effectively 'factorized' as it independently applies one-dimensional RNNs across each dimension. This approach differs from the tightly coupled multidimensional recurrences used by MDRNNs (Graves et al., 2007; Graves & Schmidhuber, 2009) and GridLSTMs (Kalchbrenner et al., 2015) and more closely resembles the approach taken by ReNet (Visin et al., 2015). Our approach allows for efficient training as we can extract features from an $M \times N$ grid in $\max(M, N)$ sequential recurrent steps rather than the $M + N$ sequential steps required for tightly coupled recurrences. Additionally, our approach enables the use of highly optimized one-dimensional RNN implementations.

Various approaches to image generation have succeeded in generating high-resolution, globally coherent images with hundreds of thousands of dimensions (Karras et al., 2017; Reed et al., 2017; Kingma & Dhariwal, 2018). The methods introduced in these works are not directly transferable to waveform generation, as they exploit spatial properties of images which are absent in one-dimensional audio signals. However, these methods are more straightforwardly applicable to two-dimensional representations such as spectrograms. Of particular relevance to our work are approaches which combine autoregressive models with multiscale modelling (van den Oord et al., 2016c; Dahl et al.,

2017; Reed et al., 2017; Menick & Kalchbrenner, 2018). Our work demonstrates that the benefits of a multiscale autoregressive model extend beyond the task of image generation, and can be used to generate high-resolution, globally coherent spectrograms.

## 8 Conclusion & Future Work

We have introduced MelNet, a generative model for spectral representations of audio. MelNet combines a highly expressive autoregressive model with a multiscale modelling scheme to generate high-resolution spectrograms with realistic structure on both local and global scales. In comparison to previous works which model time-domain signals directly, MelNet is particularly well-suited to model long-range temporal dependencies. Experiments show promising results across a diverse set of audio generation tasks.

Furthermore, we believe MelNet provides a foundation for various directions of future work. Two particularly promising directions are text-to-speech synthesis and representation learning:

- **Text-to-Speech Synthesis**: MelNet utilizes a more flexible probabilistic model than existing end-to-end text-to-speech models, making it well-suited to model expressive, multi-modal speech data.
- **Representation Learning**: MelNet is able to uncover salient structure from large quantities of unlabelled audio. Large-scale, pre-trained autoregressive models for language modelling have demonstrated significant benefits when fine-tuned for downstream tasks. Likewise, representations learned by MelNet could potentially aid downstream tasks such as speech recognition.

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

# A APPENDIX

## A.1 HYPERPARAMETERS & TRAINING DETAILS

All RNNs use LSTM cells (Hochreiter & Schmidhuber, 1997). All models are trained with RMSProp (Tieleman & Hinton, 2012) with a learning rate of $10^{-4}$ and momentum of $0.9$. The initial values for all recurrent states are trainable parameters. A single hyperparameter controls the width of the network—all hidden sizes (RNN state size, residual connections, etc.) are defined by a single value, denoted *hidden size* in table 2.

Table 2: MelNet hyperparameters.

|                          | Blizzard     | MAESTRO     | VoxCeleb2   |
|--------------------------|--------------|-------------|-------------|
| Tiers                    | 6            | 4           | 5           |
| Layers (Initial Tier)    | 12           | 16          | 16          |
| Layers (Upsampling Tiers) | 5-4-3-2-2   | 6-5-4       | 6-5-4-3     |
| Hidden Size              | 512          | 512         | 512         |
| GMM Mixture Components    | 10          | 10          | 10          |
| Batch Size               | 32           | 16          | 128         |
| Sample Rate (Hz)         | 22,050       | 22,050      | 16,000      |
| Max Sample Duration (s)  | 10           | 6           | 6           |
| Mel Channels             | 256          | 256         | 180         |
| STFT Hop Size            | 256          | 256         | 180         |
| STFT Window Size         | $6 \cdot 256$ | $6 \cdot 256$ | $6 \cdot 180$ |

## A.2 WAVENET BASELINE

The human evaluation experiments require samples from a baseline WaveNet model. For the Blizzard and VoxCeleb2 datasets, we use our own reimplementation. Our WaveNet model uses 8-bit $\mu$-law encoding and models each sample with a discrete distribution. Each model is trained for 150,000 steps. We use the Adam optimizer (Kingma & Ba, 2014) with a learning rate of 0.001 and batch size of 32. Additional hyperparameters are reported in Table 3.

Table 3: WaveNet hyperparameters.

|                            | Blizzard           | VoxCeleb2          |
|----------------------------|--------------------|--------------------|
| Sample Rate (Hz)           | 22,050             | 16,000             |
| Layers                     | 50                 | 60                 |
| Kernel Size                | 3                  | 3                  |
| Dilation (at layer $i$)    | $2^{i \bmod 10}$   | $2^{i \bmod 10}$   |
| Residual Channels          | 512                | 512                |
| Skip Channels              | 512                | 512                |
| Receptive Field (samples)  | 10,240             | 12,288             |
| Receptive Field (ms)       | 464                | 768                |
| Max Sample Duration (s)    | 2                  | 2                  |

We do not use our WaveNet implementation for human evaluation on the MAESTRO dataset. The authors that introduce this dataset provide roughly 2 minutes of audio samples on their website for both unconditional WaveNet and Wave2Midi2Wave models. We generate 50 random 10 second slices from these 2 minutes and directly use them for the human evaluations.

