# OpenReview forum: "MelNet: A Generative Model for Audio in the Frequency Domain"
_ICLR.cc/2020/Conference — Reject_

### Official Review · AnonReviewer1 · 2019-10-22
**Official Blind Review #1**

**Rating:** 3

**Review:**

This work treats 2-D spectrogram as image and uses an autoregressive models which factorizes over both time and frequency dimensions.

Detailed comments:

- MelNet is not a "fully end-to-end generative model of audio". It generates the spectrogram and relies on other algorithmic component (Griffin-Lim or gradient-based inversion) to generate raw audio.

- MelNet can model the long range structure for unconditional generation of speech, but its audio fidelity is not as good as autoregressive or non-autoregressive models on raw waveforms. The major reason is that MelNet discards the phase information which is useful for high-fidelity speech synthesis. It would be more interesting if MelNet jointly models the magnitude and phase information.

- The mixture density networks are well known. One may omit the details (or put them in Appendix) in Section 3 for space reason. Overall, the paper is clearly written, but it can be shortened in several ways.

- "making the use of 2D convolution undesirable."
It's still unclear the conv2d is useful or undesirable for modeling spectrograms in generative tasks. Even in recognition tasks, I have seen different results from different papers for different settings,  e.g., In Deep Speech 2, conv2d is useful to reduce WER in ASR.

- Missing connection in related work: previous conditional generation methods (e.g., Tacotron, Deep Voice 3) are autoregressive over time, but assume conditional independence over frequency bins.

- There is TTS experiments on the demo website, but I didn't find any details. For example, where does the conditional information (**aligned** linguistic feature) come from?

My major concern is about the usefulness of the model:

1) The unconditional speech generation is an uncommon & less useful task in general. If the task is purposely constructed, the learned representation is more useful than the generation itself (e.g., van den Oord et al. 2017). However, the authors have not demonstrated the usefulness of the learned representation for any downstream task.

2) The MelNet is autoregressive over both time and frequency. Thus, it is as slow as autoregressive waveform models at synthesis with worse audio fidelity, which make it less preferred in potential TTS applications.

**Experience Assessment:**

I have published in this field for several years.

**Review Assessment: Checking Correctness Of Derivations And Theory:**

I carefully checked the derivations and theory.

**Review Assessment: Checking Correctness Of Experiments:**

I carefully checked the experiments.

**Review Assessment: Thoroughness In Paper Reading:**

I read the paper at least twice and used my best judgement in assessing the paper.

---

> ### Author Response · Authors · 2019-11-12
> **Author Response**
>
> Thank you for the detailed review.
>
> > MelNet is not a "fully end-to-end generative model of audio". It generates the spectrogram and relies on other algorithmic component (Griffin-Lim or gradient-based inversion) to generate raw audio.
>
> We used the phrase ‘end-to-end’ to draw a distinction between MelNet and other generative models which rely on labeled intermediate representations e.g. Wave2Midi2Wave which is conditioned on symbolic music representations or text-to-speech models which use intermediate linguistic features.  Nonetheless, since the phrase is ambiguous and potentially confusing, we’ve removed it from the paper.
>
>
> > It's still unclear the conv2d is useful or undesirable for modeling spectrograms in generative tasks. Even in recognition tasks, I have seen different results from different papers for different settings,  e.g., In Deep Speech 2, conv2d is useful to reduce WER in ASR.
>
> Our phrasing on this could have been more clear.  Deep Speech 2 shows that 2d convolution is preferable to 1d convolution.  Our intention was to state that 2d convolution is in theory less appropriate than other 2d primitives such as 2d recurrence which would not assume invariance along the frequency axis.  We’ve clarified this in the revised paper.
>
>
> > Missing connection in related work: previous conditional generation methods (e.g., Tacotron, Deep Voice 3) are autoregressive over time, but assume conditional independence over frequency bins.
>
> We’ve added a paragraph in related work which references these and other related works.  We originally did not include discussion of these works since our work focuses on a broader class of generative models that are flexible enough to model arbitrary distributions of audio.   Models such as Tacotron/DV3 would not be capable of complex generation tasks such as unconditional generation for various reasons, including a) they assume conditional independence as you mentioned b) they assume unimodality (they utilize L1/L2 or other convex losses) and aim to capture a single mode of the distribution rather than capture the full variability of the distribution c) most variants of these models are deterministic.
>
>
> > There is TTS experiments on the demo website, but I didn't find any details. For example, where does the conditional information (**aligned** linguistic feature) come from?
>
> The TTS audio samples can be ignored.  The site is also used as a supplement for an extended version of the paper which also covers TTS.
>
>
> > MelNet is autoregressive over both time and frequency. Thus, it is as slow as autoregressive waveform models at synthesis with worse audio fidelity, which make it less preferred in potential TTS applications.
>
> Sampling speed is certainly one of the largest disadvantages of autoregressive models such as ours.  The original version of WaveNet also suffered from this problem but this was largely overcome following contributions in subsequent works.  We believe similar techniques can help improve the sampling speed of MelNet, but as this is a nontrivial task we believe this line of research is better suited for future work.
>
>
> > Unconditional speech generation is an uncommon & less useful task in general. If the task is purposely constructed, the learned representation is more useful than the generation itself (e.g., van den Oord et al. 2017). However, the authors have not demonstrated the usefulness of the learned representation for any downstream task.
>
> While we have not directly demonstrated the usefulness of the learned representations, we believe that the samples provide some evidence that the model has learned representations which contain useful information.  For example, when priming the state with a sequence of audio, the generated continuation preserves the characteristics of the priming sequence, suggesting that the latent characteristics of the priming sequence have been encoded into the hidden state.
>
> Regarding the usefulness of our contribution, we agree that unconditional generation is not a particularly useful task.  However, our primary goal is to present a fundamental contribution to generative models of audio which can facilitate further applications.  Unconditional image and text generation are similarly frivolous, but research in these areas lays a foundation for subsequent application-oriented research.  For example, advances in language modelling for text have led directly to improvements in translation and summarization.
>
>
> Please let us know if there are any further changes we could make for you to consider increasing your score.

---

### Official Review · AnonReviewer3 · 2019-10-23
**Official Blind Review #3**

**Rating:** 6

**Review:**

In this paper the authors present a new generative model for audio in the frequency domain to capture better the global structure of the signal. For this, they use an autoregressive  procedure combined with a multiscale generative model for two-dimensional time-frequency visual representation (STFT spectrogram). The proposed method is tested across a diverse set of audio generation tasks

Overall, The idea of generating audio from 2D spectrogram is original but in my point of view the use of STFT is not appropriate in this context, especially with its lossy criterion.

Given the clarifications and the author’s responses below, I increased the score from 3 to 6.



Detailed comments:

Pro:

Mitigate the problem of generating signal using only local dependencies (on a narrow time scale) and this by capturing high level dependency that emerges on larger timescale (several seconds) using spectrogram.



Cons:

(1)The use of  STFT is not justified why not wavelet spectrogram to capture both scale and time?
(2)It is still confusing how the use of high resolution spectrogram improve the lossy representation?
(3)If you increase the STFT hope size you come back to the main problem that you are trying to resolve (i.e,  the bias towards capturing local dependencies)


**Experience Assessment:**

I have read many papers in this area.

**Review Assessment: Checking Correctness Of Derivations And Theory:**

I carefully checked the derivations and theory.

**Review Assessment: Checking Correctness Of Experiments:**

I assessed the sensibility of the experiments.

**Review Assessment: Thoroughness In Paper Reading:**

I read the paper at least twice and used my best judgement in assessing the paper.

---

> ### Author Response · Authors · 2019-11-12
> **Author Response**
>
> Thank you for the comments and suggestions.
>
> > The use of  STFT is not justified why not wavelet spectrogram to capture both scale and time?
>
> Thanks for suggesting the use of wavelet-based representations. Our primary aim was to demonstrate that autoregressive models of time-frequency representations offer advantages (e.g. better modelling of long-range structure) in comparison to existing autoregressive models of time-domain waveforms.  For this reason, our study does not focus on variations within the class of time-frequency representations and we instead opted (somewhat arbitrarily) to use an STFT-based representation.  While beyond the scope of our paper, we believe that exploring alternative time-frequency representations would be interesting future work.
>
>
> >It is still confusing how the use of high resolution spectrogram improve the lossy representation?
>
> The STFT itself is invertible, but we use mel-spectrograms which discard information through a) the removal of phase and b) the compression of the frequency axis via the mel transform.   Ideally we’d like to minimize the amount of information removed from the STFT by these transformations.  Intuitively, we can minimize the information loss due to the mel transform by utilizing a larger number of frequency bins.  Similarly, to minimize the information loss due to the removal of phase, we can use a smaller STFT hop size which eases the task of phase estimation by algorithms such as Griffin-Lim.  Empirically, this can be validated by inverting spectrograms at various resolutions and observing the that higher resolution spectrograms can be inverted to higher fidelity audio.
>
>
> > If you increase the STFT hope size you come back to the main problem that you are trying to resolve (i.e,  the bias towards capturing local dependencies)
>
> Correct, and this is precisely the motivation for the multiscale approach described in the paper.  By separately modeling the spectrogram at different scales, it is possible to largely decouple the tasks of modeling high-level and low-level structure.
>
>
> > Given the clarifications and the author’s responses, I would be willing to increase the score.
>
> Let us know if there are any further concerns you would like us to address.

---

### Official Review · AnonReviewer2 · 2019-10-23
**Official Blind Review #2**

**Rating:** 8

**Review:**

The authors introduce MelNet, an autoregressive model of Mel-frequency scaled spectrograms. They convert audio into high resolution spectrograms to reduce the audio artifacts introduced by inverting spectrograms (here they use gradient-based inversion over Griffin-Lim). To improve modeling of long term dependencies, they perform multi-scale splitting of the spectrograms and maximize the likelihood at each scale (avoiding dominance of noise at higher resolutions). They condition generation at finer scales from coarser scales, enabling sampling through an ancestral process. The authors also highlight the difference between temporal and frequency dimensions, creating different conditioning stacks for the past in time vs. the "past" in frequency (lower frequencies), and mixing conditioning between the two stacks through layers of the network. Multilayer RNNs are used throughout the network and external conditioning is incorporated at the input.

The challenge the authors are attempting to address is modeling of audio structure on both long and short timescales. As the authors demonstrate with strong baselines, WaveNet models, while superior on fine-scale fidelity, fail to capture dynamics more than a couple hundred milliseconds. The experiments demonstrate improvements on state-of-the-art for unconditional generation on text-to-speech datasets (generating coherent words and phrases) and the MAESTRO piano dataset (generating sections with consistent dynamics/timing/motifs). The continuations of primed examples in both domains are particularly impressive qualitatively, as they maintain much of the character of the priming sample. Ablation experiments qualitatively demonstrate the importance of multi-scale modeling for unconditional generation. Human listener studies support the claims made from qualitative evaluation of long term structure.

This paper should be accepted because it represents a non-trivial adaptation of autoregressive modeling to handle multi-scale structure in audio. The baselines comparisons and strong, and experiments validate the claims of the paper.

That said, several things could be done to improve the clarity and significance of the paper.

* While the network architecture is described in detail and some figures, the full network structure itself is non-trivial and still somewhat opaque from the plain text description. A schematic diagram of the full network architecture, even in the appendix, could help clarify how many layers are present connecting each component of the model, which would improve reproducibility.

* The paper is a bit thin on metrics. Human listening studies compare long-term structure, but not short-scale fidelity. For TTS, there are clear artifacts from the spectrogram inversion process. Mean opinion scores on conditional samples could help to quantify the importance of each element of the network for audio quality. For instance, how does MOS compare between Griffin-Lim MelNet, Gradient Inversion MelNet, and WaveNet? How does MelNet compare to Linear scaled spectrograms?

* Generating MelSpectrograms to model long-term structure is a fairly established technique, most notably employed by all of the Tacotron variants (https://google.github.io/tacotron/). These models are perhaps a more appropriate comparison for MelNet in many ways, and opt for spectrogram inversion by smaller WaveRNN models. One of the claims of the paper is that it is important to model the fine-scale structure of spectrograms, but it is not clear if that really is the case. A proper comparison to Tacotron models (where spectrograms are generated at the same resolution / the same inversion methods are used) would help clarify the importance of end-2-end training, vs. the learned inversion approach.



**Experience Assessment:**

I have published in this field for several years.

**Review Assessment: Checking Correctness Of Derivations And Theory:**

N/A

**Review Assessment: Checking Correctness Of Experiments:**

I carefully checked the experiments.

**Review Assessment: Thoroughness In Paper Reading:**

I read the paper thoroughly.

---

> ### Author Response · Authors · 2019-11-12
> **Author Response**
>
> Thank you for the comments and suggestions.
>
> > While the network architecture is described in detail and some figures, the full network structure itself is non-trivial and still somewhat opaque from the plain text description. A schematic diagram of the full network architecture, even in the appendix, could help clarify how many layers are present connecting each component of the model, which would improve reproducibility.
>
> We spent a fair amount of time discussing how to show the architecture schematically, but ultimately did not come up with anything particularly satisfying.  Most of the intricacy lies in the implementation of the multi-dimensional recurrence and the connectivity between the time and frequency stacks, so we focused on including equations and figures which provide clarity in these areas.
>
>
> > The paper is a bit thin on metrics. Human listening studies compare long-term structure, but not short-scale fidelity. For TTS, there are clear artifacts from the spectrogram inversion process. Mean opinion scores on conditional samples could help to quantify the importance of each element of the network for audio quality. For instance, how does MOS compare between Griffin-Lim MelNet, Gradient Inversion MelNet, and WaveNet? How does MelNet compare to Linear scaled spectrograms?
>
> We generally agree with your comments here.  As we don’t provide any experiments regarding short-term fidelity, we made sure to restrict our claims to state that the benefit of MelNet relates to its ability to capture long-range structure.  In cases where audio fidelity is paramount, time-domain models could be used to invert spectrograms generated by MelNet.
>
>
> >  Generating MelSpectrograms to model long-term structure is a fairly established technique, most notably employed by all of the Tacotron variants (https://google.github.io/tacotron/). These models are perhaps a more appropriate comparison for MelNet in many ways, and opt for spectrogram inversion by smaller WaveRNN models. One of the claims of the paper is that it is important to model the fine-scale structure of spectrograms, but it is not clear if that really is the case. A proper comparison to Tacotron models (where spectrograms are generated at the same resolution / the same inversion methods are used) would help clarify the importance of end-2-end training, vs. the learned inversion approach.
>
> The main distinction between MelNet and models such as Tacotron is that MelNet utilizes a much more flexible probabilistic model that allows it to be applied to unconditional generation tasks such as music generation.  The experiments in this paper are restricted to unconditional audio generation, so direct comparison with models such as Tacotron would not be possible.  We’ve revised the paper to include a discussion of existing TTS models.

---

### Decision · Program_Chairs · 2019-12-19

**Decision:**

Reject

**Comment:**

The paper proposed an autoregressive model with a multiscale generative representation of the spectrograms to better modeling the long term dependencies in audio signals. The techniques developed in the paper are novel and interesting. The main concern is the validation of the method. The paper presented some human listening studies to compare long-term structure on unconditional samples, which as also mentioned by reviewers are not particularly useful. Including justifications on the usefulness of the learned representation for any downstream task would make the work much more solid.